

# 1  Impact of buildings on surface solar radiation over urban
# 2  Beijing

**B. Zhao[1], K. N. Liou[1], Y. Gu[1], C. He[1], W. L. Lee[2], X. Chang[3], Q. B. Li[1], S. X. Wang[3,**
**4], H. R. Tseng[1], L. R. Leung[5], J. M. Hao[3, 4]**
[1] Joint Institute for Regional Earth System Science and Engineering and Department of
Atmospheric and Oceanic Sciences, University of California, Los Angeles, CA 90095, USA
[2] Research Center for Environmental Changes, Academia Sinica, Taipei, Taiwan
[3] State Key Joint Laboratory of Environment Simulation and Pollution Control, School of
Environment, Tsinghua University, Beijing 100084, China
[4] State Environmental Protection Key Laboratory of Sources and Control of Air Pollution
Complex, Beijing 100084, China
[5] Pacific Northwest National Laboratory, Richland, WA 99352, USA
Correspondence to: B. Zhao [zhaob1206@gmail.com]
**Abstract.**
The rugged surface of an urban area due to varying buildings can interact with solar beams
and affect both the magnitude and spatiotemporal distribution of surface solar fluxes. Here we
systematically examine the impact of buildings on downward surface solar fluxes over urban
Beijing by using a 3-D radiation parameterization that accounts for 3-D building structures
versus the conventional plane-parallel scheme. We find that the resulting downward surface
solar flux deviations between the 3-D and the plane-parallel schemes are generally ±1–
10 W m$^{-2}$ at 800-m grid resolution and within ±1 W m$^{-2}$ at 4-km resolution. Pairs of positive-
negative flux deviations on different sides of buildings are resolved at 800-m resolution, while
they offset each other at 4-km resolution. Flux deviations from the unobstructed horizontal
surface at 4-km resolution are positive around noon but negative in the early morning and late
afternoon. The corresponding deviations at 800-m resolution, in contrast, show diurnal
variations that are strongly dependent on the location of the grids relative to the buildings.
Both the magnitude and spatiotemporal variations of flux deviations are largely dominated by
the direct flux. Furthermore, we find that flux deviations can potentially be an order of
magnitude larger by using a finer grid resolution. Atmospheric aerosols can reduce the





magnitude of downward surface solar flux deviations by 10–65%, while the surface albedo
generally has a rather moderate impact on flux deviations. The results imply that the effect of
buildings on downward surface solar fluxes may not be critically significant in mesoscale
atmospheric models with a grid resolution of 4 km or coarser. However, the effect can play a
crucial role in meso-urban atmospheric models as well as microscale urban dispersion models
with resolutions of 1 m – 1 km.
**1   Introduction**
The spatial orientation and inhomogeneous features of the earth's surface interact with direct
and diffuse solar beams in an intricate manner (Liou et al., 2013). In particular, the complex
and rugged surface of an urban area due to varying buildings can interact with solar beams
and affect both the magnitude and spatiotemporal distribution of surface solar fluxes. The
distribution of solar fluxes can significantly modulate surface heating and moistening,
evapotranspiration, land-atmosphere interaction, boundary layer, and air pollutant dispersion
(Lee et al., 2011; Gu et al., 2012). It is very difficult to accurately quantify the surface solar
flux distribution in view of the complexity of spatial orientation and surface optical properties,
especially over urban areas.

18       Several approaches with varying degrees of sophistication have been developed to evaluate

solar fluxes at rugged surface (Dozier and Frew, 1990; Dubayah et al., 1990; Chen et al., 2006;
Essery and Marks, 2007; Lai et al., 2010). Among these approaches, the 3-D Monte Carlo
photon tracing approach gives the most physically-representative radiative transfer
calculations for an environment with complex 3-D topography. Chen et al. (2006) and Liou et
al. (2007) developed a Monte Carlo program and found that the domain-average downward
surface solar fluxes with rugged topography deviate from the unobstructed horizontal surface
by 10–50 $W\,m^{-2}$ over the Tibetan Plateau and can be as large as 600 $W\,m^{-2}$ locally over
shaded areas. The 3-D Monte Carlo approach has also been used to evaluate interactions
between solar beams and other irregular surfaces, such as wind-blown sea surfaces and plant
canopies (Preisendorfer and Mobley, 1986; Iwabuchi and Kobayashi, 2008; Mayer et al.,
2010). However, a drawback of the 3-D Monte Carlo photon tracing approach is the enormous
computational burden. To overcome this drawback, Lee et al. (2011, 2013) developed a
parameterization of downward solar fluxes associated with topographic information based on
3-D Monte Carlo simulations. The parameterization was subsequently implemented in



regional and global weather and climate models (Liou et al., 2013; Lee et al., 2015; Gu et al.,
2012) in which the effects of 3-D mountainous topography on sensible and latent heat fluxes,
surface hydrology, and cloud properties have been investigated and evaluated.
With the objective to improve the urban representation in land-surface schemes that has
been used in numerical models, a number of urban energy balance models (or urban canopy
models) have been developed, as reviewed by Grimmond et al. (2010, 2011). Some of these
models have considered a building's shading effect and the reflectance of solar beams by
building walls (Kusaka et al., 2001; Kusaka and Kimura, 2004; Kondo et al., 2005; Oleson et
al., 2008). However, these models have at least two drawbacks. First, the 3-D radiative
transfer was calculated based on simplified, evenly spaced buildings of the same height, rather
than "real" buildings. Second, the diffuse, diffuse-reflected, and coupled fluxes (e.g., multiple
reflections) were often oversimplified, resulting in noticeable errors due to the distinct
features of the different flux components. A systematic evaluation and physical understanding
of the 3-D building effect on surface solar radiation over urban areas is imperative.
In this study, we investigate the impact of buildings on downward surface solar fluxes over
urban Beijing, the capital and one of the largest megacities in China. The evaluation is
conducted using the 3-D radiation parameterization developed by Lee et al. (2013) coupled
with the Fu-Liou-Gu (FLG) plane-parallel radiative transfer scheme (Fu and Liou, 1992; Gu
et al., 2003; Gu et al., 2006). In Section 2, we describe the parameterization of 3-D
topography effect on downward solar fluxes and its application over urban Beijing. In Section
3, we investigate the magnitude and spatiotemporal variation of deviations in downward
surface solar fluxes induced by buildings and evaluate the effect of key factors by means of
sensitivity simulations. Conclusions and implications are given in Section 4.

## 2  Methodology and data source

### 2.1  Parameterization of the 3-D topography effect on downward surface solar fluxes

In order to evaluate the impact of buildings on downward surface solar radiation, we apply the
3-D radiation parameterization over rugged surface developed by Lee et al. (2013). Below are
key points of the parameterization. Note that we focus exclusively on "downward" solar
fluxes in this study.



Solar radiative fluxes can be categorized into five components according to photon path: (1)
direct flux ($F_{dir}$) is composed of photons hitting the ground directly from the sun without
encountering scattering or reflection; (2) diffuse flux ($F_{dif}$) contains photons experiencing
single or multiple scattering by air molecules, but does not encounter surface reflection; (3)
direct-reflected flux ($F_{rdir}$) is comprised of unscattered photons reflected by nearby terrain;
(4) diffuse reflected flux ($F_{rdif}$) means that photon is first scattered by air molecules and then
reflected by nearby terrain; and (5) coupled flux ($F_{coup}$) represents photons that after being
reflected by the surface, encounter scattering and/or one or more additional surface
reflections.
Conventional plane-parallel radiative transfer schemes have already been developed to
calculate solar fluxes on a horizontal surface, so the purpose of the 3-D radiation
parameterization is to produce relative deviations of these five flux components from those of
an unobstructed horizontal surface. On the basis of 3-D Monte Carlo photon tracing
simulations, Lee et al. (2011, 2013) utilized a multiple linear regression technique to establish
the relationship between deviations in solar fluxes (response variables) and subgrid scale
topographic information (independent variables). The Shuttle Radar Topography Mission
(SRTM) topography data (Jarvis et al., 2008) at a resolution of 3 arc-second (about 90 m)
were used to perform 3-D Monte Carlo simulations for many $10\times10$ km$^2$ rugged domains in
the Sierra Nevada Mountain area, which were subsequently used to develop regression
parameterization. Although the parameterization was developed in the Sierra Nevada area, it
is applicable to other regions because it is topographic parameter-dependent rather than
location-dependent. The regression equations for flux deviations in clear-sky condition can be
expressed by
$$
\begin{pmatrix} F_{dir}^{'} \\ F_{dif}^{'} \\ F_{rdir}^{'} \\ F_{rdif}^{'} \\ F_{coup}^{'} \end{pmatrix} = \begin{pmatrix} a_1 \\ a_2 \\ a_3 \\ a_4 \\ a_5 \end{pmatrix} + \begin{pmatrix} b_{11} & b_{12} & 0 & 0 \\ b_{21} & b_{22} & 0 & b_{24} \\ 0 & b_{32} & b_{33} & 0 \\ 0 & b_{42} & b_{43} & 0 \\ b_{51} & b_{52} & b_{53} & 0 \end{pmatrix} \begin{pmatrix} \langle \tilde{\mu}_i \rangle \\ \langle \tilde{V}_d \rangle \\ \langle \tilde{C}_t \rangle \\ \sigma(h) \end{pmatrix},
\tag{1}
$$

where $F_i^{'}$ is the relative deviation of each flux component, i = $dir$, $dif$, $rdir$, $rdif$, and $coup$. $a_i$
is the interception, $b_{ij}$ is the regression coefficient for a specific independent variable. $\tilde{\mu}_i$ is





the cosine of the solar zenith angle normalized by the cosine of the slope, $\tilde{V}_d$ is the sky view
factor normalized by the cosine of the slope, $\tilde{C}_t$ is the terrain configuration factor normalized
by the cosine of the slope, $\sigma(h)$ is the standard deviation of elevation, and angle brackets
denote the spatial mean of the variable within a $10 \times 10$ km$^2$ domain. Lee et al. (2013)
demonstrated that the flux components predicted by these regression equations agree well
with those directly calculated from Monte Carlo simulations.
## 2.2   Application of the 3-D radiation parameterization to urban Beijing
We apply the parameterization described above to Beijing, a megacity with numerous
buildings, many of which are skyscrapers. Two domains with different sizes and resolutions
are used (Fig. 1). Domain 1 covers urban and suburban Beijing at a grid resolution of 4 km,
which is a commonly used resolution in mesoscale atmospheric models. The Xishan mountain
is located in the northwestern part of the domain, serving as a comparison of the 3-D
topography effect over mountainous and urban areas. The rest of the domain is characterized
by plains with typical urban landscape (e.g., buildings and roads). Domain 2 covers the urban
center of Beijing at 800-m resolution, corresponding to the typical resolution of meso-urban
models.
Following Lee et al. (2013), we adopt the topography data at a resolution of 3 arc-second
(about 90 m) from SRTM, and calculate average topographical parameters for each 4 km or
800 m grid in the simulation domains. Figure 1 (right panel) shows that major buildings are
resolved in the 90 m topography data. The SRTM data is for the year 2000. We note that
urban development in Beijing has expanded greatly since 2000, far beyond what is
represented in the SRTM data. This study aims to assess the potential magnitude of the effect
of buildings on solar fluxes; the SRTM data meet the need considering that there were already
numerous buildings in Beijing in 2000.
The 3-D radiation parameterization was originally developed for $10 \times 10$ km$^2$ grids. Lee et al.
(2011, 2013) demonstrated its compatibility across various resolutions. Theoretically it should
be applicable for a grid resolution as fine as 800 m since an $800 \times 800$ m$^2$ grid still comprises a
large quantity of 90 m pixels. Here we further evaluate the compatibility associated with
resolutions by comparing the flux deviations in each $4 \times 4$ km$^2$ grid calculated directly from the
3-D parameterization and those from the summation of all $800 \times 800$ m$^2$ grids. We find the
biases between the two are within $\pm 0.025$ W m$^{-2}$, indicating a reasonable compatibility





between different grid resolutions. The calculation method and subsequent results are
described in detail in the Supplementary Material.
The 3-D radiation parameterization is used in conjunction with the FLG plane-parallel
radiation scheme (Fu and Liou, 1992; Gu et al., 2003; Gu et al., 2006; Gu et al., 2010), which
calculates solar fluxes on flat surfaces. The FLG scheme combines the delta-four-stream
approximation for solar flux calculations with the delta-two/four-stream approximation for
infrared flux calculations to assure both accuracy and efficiency. The solar (0–5μm) and
infrared (5–50μm) spectra are divided into 6 and 12 bands, respectively, within which the
correlated k-distribution method is used to sort gaseous absorption lines. The single-scattering
properties of 18 aerosol types are parameterized by employing the Optical Properties of
Aerosols and Clouds (OPAC) database.
The meteorological and chemical variables (i.e., air temperature, surface temperature,
pressure, humidity, surface albedo, ozone concentrations, and aerosol optical depth) used in
the FLG scheme are derived from a simulation of the Weather Research and Forecasting
model (WRF, version 3.3)/Community Multi-scale Air Quality model (CMAQ, version 5.0.2).
The conversion of vertically resolved aerosol mass concentrations to aerosol optical depth
follows Heald (2010) and Martin and Heald (2010). For the WRF/CMAQ simulation, we
apply one-way, triple nesting domains with resolutions of 36 km, 12 km, and 4 km,
respectively (Fig. S1). The simulated meteorological parameters and concentrations of fine
particles (PM$_{2.5}$) and their chemical components are in reasonable agreement with
observations (Table S2, Fig. S2). The configuration of WRF/CMAQ and its evaluation against
observations are described in detail in the Supplementary Material. The meteorological and
chemical variables of Domain 1 (4-km resolution) are taken from the WRF/CMAQ simulation
directly, while the varibles in Domain 2 (800-m resolution) are assumed to be the same as
their corresponding values at the 4 km grids.
The 3-D radiative transfer calculations are for January 1$^{st}$, April 1$^{st}$, July 1$^{st}$, and October 1$^{st}$,
2012, representing four seasons. Within each day, the calculation is done every hour starting
from 0:00, Beijing Time (BT). To avoid the fluctuation of atmospheric profiles, we conduct
the WRF/CMAQ simulations for four months (January, April, July, and October) and use
monthly average meteorological and chemical variables for each of the 24 hours in the 3-D
radiative transfer calculations. For example, for the simulation of January 1$^{st}$ 0:00 BT, we use
the average temperature at 0:00 BT of each day in January.



We conduct radiative transfer computation primarily for clear-sky condition without
aerosols, for which the 3-D radiation parameterization was developed. We also incorporate
aerosols for a sensitivity scenario (see Section 3.3.1). In the presence of aerosols, regression
equations for $F'_{dir}$ and $F'_{rdir}$ can be directly applied because these two components do not
encounter scattering. As for $F'_{dif}$, $F'_{rdif}$, and $F'_{coup}$, the parameterization provides a first-order
estimate (Lee et al., 2013; Lee et al., 2011). Considering that the direct flux usually dominates
over other components (Chen et al., 2006; Lee et al., 2011), the parameterization is likely
applicable in an environment with a large aerosol loading.
**3    Results and discussion**
**3.1    Deviations in solar fluxes from horizontal surface**
We calculate surface solar fluxes at rugged city surface by employing the 3-D radiation
parameterization coupled with the FLG plane-parallel scheme. Surface solar flux deviations
between the 3-D radiation parameterization and plane-parallel scheme represent the effect of
buildings. Figure 2 (top three rows) shows hourly flux deviations at selected times (7:00,
12:00, and 17:00 BT) on April 1$^{st}$ in clear-sky condition without aerosols. Figure 3 depicts
daily average flux deviations for four simulation days (January 1$^{st}$, April 1$^{st}$, July 1$^{st}$, and
October 1$^{st}$). For Domain 1 (4-km resolution), a striking feature is that deviations over urban
areas are remarkably smaller than those over mountainous areas. Both hourly and daily
average deviations over urban areas are generally within ±1 W m$^{-2}$. In contrast, hourly/daily
average deviations over mountainous areas are on the order of ±10–70 W m$^{-2}$, except for July
when daily average deviations are generally within 10 W m$^{-2}$. The maximum local deviations
can be up to ±100 W m$^{-2}$. In Domain 2 (800-m resolution), both the magnitude and the spatial
pattern of deviations differ greatly from Domain 1. Flux deviations usually range between ±1–
10 W m$^{-2}$. The magnitude of flux deviations has a significant seasonal variation associated
with the position of the sun in different seasons. For example, daily average flux deviations
are within ±10 W m$^{-2}$, ±6 W m$^{-2}$, and ±1 W m$^{-2}$ in January, April/October, and July,
respectively. Smaller daily average deviations in July are attributable to the smaller shading
effect at the north-south direction as the sun is close to its zenith at noon. In addition, the fine
structure of positive-negative pairs on southern-northern or eastern-western sides of buildings
is resolved in Domain 2. This phenomenon is especially pronounced when we compare flux
deviations at 7:00 BT and 17:00 BT. Many grids show opposite-sign flux deviations at these





two times, implying that they are located on the opposite side of buildings. The spatial pattern
comprising of positive-negative pairs is somewhat similar to that of mountainous areas in
Domain 1. By comparing Domain 1 and Domain 2, we conclude that flux deviations from the
flat surface over urban areas are quite sensitive to grid resolution. The magnitude of
deviations is small at a coarse resolution such as 4 km, because of the offset of postive and
negative deviations.
We futher analyze the diurnal variation of flux deviations from the horizontal surface, as
shown in Fig. 4. To facilitate the analysis, we select a typical mountainous area (defined as
rectangle A in Fig. 1) and a typical urban area (defined as rectangle B in Fig. 1) in Domain 1,
as well as a typical urban area (defined as rectangle C in Fig. 1) in Domain 2. Flux deviations
in the typical urban area defined in Domain 1 (Fig. 4b) are positive during 6–7 hours around
noon with peaks occuring at noon, while they are negtive in the early morning and late
afternoon. This diurnal pattern persists on all simulation days. At noon, buildings generally
receive more solar energy than a flat surface due to a larger surface area facing the sun,
whereas negative deviations in the early morning and late afternoon are primarily induced by
larger shading areas. The diurnal pattern over the typical urban area defined in Domain 2 (Fig.
4c) substantially differs from the preceding pattern such that flux deviations are positive in the
morning and negative in the afternoon. Figure 1 shows that these grids are mostly located in
the eastern side of the buildings rather than the western side. In this case, the eastern side
faces the sun in the morning, receiving more solar fluxes than its horizontal surface
counterpart. In the afternoon, the eastern side is shaded by the buildings to substantially block
solar beam. We note that the diurnal variation of grids in Domain 2 is a strong function of
their relative locations to the buildings. For example, the diurnal pattern is exactly opposite
for a grid containing more buildings' western side. Furthermore, it is noticeable that the
diurnal pattern of the typical urban area defined in Domain 2 highly assembles that of the
typical mountainous area defined in Domain 1 (Fig. 4a), which is located on the eastern side
of the Xishan mountain. This reveals the similarity between buildings and mountains in terms
of their impacts on surface radiation, though they are associated with different spatial scales –
4 km or more for mountains (Liou et al., 2013; Lee et al., 2013), and 800 m or less for
buildings.



## 3.2 Contribution of individual flux components to flux deviations

We quantify the contribution of individual flux components to surface solar flux deviations between 3-D and plane-parallel in order to gain a deeper understanding of the effect of buildings on solar flux distributions. Figure 5 shows the contribution of individual components to flux deviations on April 1$^{st}$ in the three typical areas defined in the last section, while Fig. S3 depicts the corresponding contributions on four simulation days (January 1$^{st}$, April 1$^{st}$, July 1$^{st}$, and October 1$^{st}$) in the typical urban area defined in Domain 1. For the other two typical areas, only April 1$^{st}$ is shown because the other simulation days present very similar patterns. As described in Section 2.1, solar fluxes are physically categorized into five components, including direct flux, diffuse flux, direct-reflected flux, diffuse-reflected flux, and coupled flux. In Fig. 5, diffuse and coupled fluxes are merged together, considering that the coupled flux is usually negligible and that these two components are treated together in the plane-parallel scheme. A striking pattern is that the direct flux largely dominates deviations from the unobstructed horizontal surface over both urban and mountainous areas. The diurnal variation of direct flux is very similar to that of the total flux, which has been illustrated in detail in the last section. In general, deviations in diffuse flux (plus coupled flux) are negative over both urban and mountainous areas since sky view factors are less than 1.0 in street canyons or valleys. Their magnitude is generally between -0.03 W m$^{-2}$ and -0.10 W m$^{-2}$ in typical urban areas in Domain 1 (Fig. 5b, Fig. S3) and between -0.10 W m$^{-2}$ and -0.25 W m$^{-2}$ in typical urban areas in Domain 2 (Fig. 5c), both peaking at noon. Deviations in direct-reflected and diffuse-reflected fluxes are always positive because these two components do not exist on unobstructed horizontal surfaces. The magnitude of direct-reflected flux ranges between 0.01–0.20 W m$^{-2}$ in typical urban areas (both Domain 1 and Domain 2), with peaks occurring at summer noon. Figure S3 shows that deviations in the direct-reflected flux can exceed those of the direct flux for a few hours around summer noon. The magnitude of diffuse-reflected flux is always negligible compared with the components described above.

## 3.3 Sensitivity analysis

### 3.3.1 Effect of aerosols on flux deviations

In preceding discussions, we focused on the effect of buildings in clear-sky condition without aerosols. Atmospheric aerosols can potentially alter the transfer of solar radiation. As described in Section 2.2, although the 3-D radiation parameterization was developed in clear-



sky condition without aerosols, regression equations for $F'_{dir}$ and $F'_{rdir}$ can be directly applied
to aerosol contaminated environment, while those for $F'_{dif}$, $F'_{rdif}$, and $F'_{coup}$ can provide a first-
order estimate. Figure 2 shows hourly flux deviations between 3-D and plane-parallel at
selected times (7:00, 12:00, and 17:00 BT) on April 1[st] with and without aerosols. The results
on the other simulation days (January 1[st], July 1[st], and October 1[st]) are quite similar, and thus
are now shown. In general, the inclusion of aerosols reduces the magnitude of surface flux
deviations without changing the spatial pattern. This can be explained by the attenuation of
total solar fluxes by aerosols across the domain. Over the urban center (Domain 2), aerosols
reduce the magnitude of daily average deviations by about 15–30%. The reduction ratios are
significantly higher in the early morning and late afternoon (40–65%) than at noon (10–25%),
mainly due to higher aerosol optical depths in the early morning/late afternoon. In this study,
interactions between buildings and aerosols are not considered in the simulation. For example,
photons reflected by buildings can further be scattered/absorbed by aerosols, and vice versa.
Given that diffuse-reflected and coupled fluxes are much smaller than direct flux, the resulting
errors should be minor. The 3-D Monte Carlo photon tracing program is needed in order to
achieve a more accurate evaluation of the effect of aerosols on flux deviations.
### 3.3.2   Sensitivity of flux deviations to spatial resolutions
As demonstrated in Section 3.1, the magnitude of flux deviations from the flat surface is quite
sensitive to spatial resolutions. Over urban areas, hourly deviations are $\pm 1$–10 W m$^{-2}$ at 800-m
resolution and within $\pm 1$ W m$^{-2}$ at 4-km resolution. The smaller values in coarser grids can be
explained by the compensation effect of positive and negative deviations on the opposite side
of buildings. Judging from the right panel of Fig. 1, an $800 \times 800$ m$^2$ grid still covers quite a
few buildings, which motivates us to explore the potential effect of buildings at even finer
resolutions. As a test case, we present a rough estimate of flux deviations at a 3 arc-second
(about 90 m) resolution (shown in Fig. 6) by applying the 3-D radiation parameterization to 3
arc-second topography data derived from SRTM. Theoretically, the parameterization may not
be applicable to a spatial resolution less than about 1 km with acceptable accuracy.
Nevertheless, it suffices to provide an initial estimate for flux deviations, though results must
be interpreted with care. Of course, a more accurate estimation should be made using the
Monte Carlo method in future studies. Figure 6 shows that hourly deviations in 90 m grids are
generally between $\pm 5$–50 W m$^{-2}$, and the maximum local deviations can reach about $\pm 100$





W m$^{-2}$. This is notably higher than flux deviations at 800-m resolution. These results highlight
the potential importance of 3-D building effects on the microscale modeling with resolutions
of 1–100 m (e.g., urban dispersion models), which requires further studies.
### 3.3.3   Sensitivity of flux deviations to the surface albedo
The surface albedo used in the 3-D radiation parameterization was directly derived from
WRF/CMAQ simulation results, which ranges between 0.15–0.20 and represents the typical
surface albedo of urban areas. However, there is a wide variety of roofing materials with
distinct albedos (Prado and Ferreira, 2005). One geoengineering proposal to ameliorate the
effect of urban heat island was to use reflective roofing material or to paint existing roofs
white (Jacobson and Ten Hoeve, 2012). There are also increasing numbers of buildings with
glass surfaces. To evaluate the potential effect of amplified surface albedo on flux deviations
from the horizontal surface, we design three sensitivity cases in which domain-wide surface
albedo was uniformly increased to 0.35, 0.50, and 0.65. Figure 7 shows simulated surface
solar flux deviations in a typical urban area in Domain 1 (defined as rectangle B in Fig. 1) as a
function of surface albedo. We focus on urban areas in Domain 1 (4-km resolution) because it
is the region where the largest relative contribution of the reflected flux is identified (see Fig.
5), implying a potentially large sensitivity to surface albedo. Figure 7 shows a moderate
impact of surface albedo on flux deviations during the day. The largest sensitivity occurs at
summer noon, at which a large albedo of 0.65 can amplify flux deviations from 0.1–0.4 W m$^{-2}$
to about 0.6 W m$^{-2}$. Compared with the case of a 4-km resolution, the change in surface
albedo results in a much smaller relative change in flux deviations at 800-m resolution,
because the relative contribution of the reflected flux is smaller at 800-m resolution (see Fig.

23   5).

### 3.4   Implications for atmospheric studies
The present results have important implication for future studies. Deviations in surface solar
fluxes are within 1 W m$^{-2}$ at a 4 km or coarser resolution due to the offset of positive and
negative flux deviations, therefore the effect of buildings may not be critically significant in
mesoscale atmospheric models. Nevertheless, the effect can not be neglected if there is a
substantially inhomogeneous subgrid-scale distribution of plants, accumulated snow, and
building/road materials, etc.; in this case, subgrid-scale flux deviations may result in biased
evapotranspiration, snowmelting, and heat fluxes, etc. For meso-urban models with a typical





resolution of about 1 km (e.g., urbanized MM5 model, uMM5; Taha et al., 2008), the 3-D building effects become quite significant (about $\pm 1$–10 W m$^{-2}$). The parameterization used in this study can be readily incorporated in these models to account for 3-D building effects. As for computational fluid dynamics models (e.g., FLUENT) and urban dispersion models (e.g., Atmospheric Dispersion Modelling System, ADMS) with resolutions of 1–100 m, this study implies that flux deviations induced by buildings might be up to $\pm 100$ W m$^{-2}$. The large flux deviations can significantly alter local energy balance, and thus affects the spatial distribution of temperature and small-scale flows around buildings and/or through street canyons. Therefore, the 3-D building effects on solar fluxes can play a crucial role in numerical simulation of urban meteorology and air pollutant dispersion. The present 3-D radiation parameterization may not be applicable to 1–100 m resolutions. As such, a more physically-based approach directly using an appropriate 3-D Monte Carlo photon tracing program will be needed to account for 3-D building effects more precisely. Also, topography data such as the recently released SRTM datasets at a resolution of 1 arc-second (about 30 m) may also be useful for the study of 3-D building effects.

## 4    Conclusions

In this study, we systematically evaluated the impact of buildings on surface solar fluxes over urban Beijing using the 3-D radiation parameterization developed in our previous study in connection with the FLG radiative transfer scheme. The evaluation was conducted in two simulation domains with grid resolutions of 4 km and 800 m, representing typical resolutions for mesoscale and meso-urban models, respectively.

Over urban Beijing, deviations in surface solar fluxes between the 3-D radiation parameterization and the plane-parallel scheme are generally $\pm 1$–10 W m$^{-2}$ at 800-m resolution and within $\pm 1$ W m$^{-2}$ at 4-km resolution. Pairs of positive-negative flux deviations on different sides of buildings are resolved at 800-m resolution, while they offset each other at 4-km resolution. Deviations in surface solar fluxes over urban areas are considerably smaller than those over mountainous areas using preceding grid resolutions.

Flux deviations over urban areas are positive around noon but negative in the early morning and late afternoon at 4-km resolution. The corresponding deviations at 800-m resolution, in contrast, show diurnal variations that are strongly dependent on the grids' relative locations to buildings. Both the magnitude and spatiotemporal variations of flux deviations are largely dominated by the direct flux.



With a series of sensitivity simulations, we show that atmospheric aerosols reduce the magnitude of surface flux deviations by 10–65% without changing the spatial pattern. Simulated deviations in surface fluxes are very sensitive to spatial resolution. They can potentially reach up to ±100 W m$^{-2}$ at a high resolution of about 90 m. The surface albedo has a moderate impact on flux deviations during the day, while the impact can be substantial at summer noon.

This study implies that the effect of buildings on surface solar fluxes may not be critically important in mesoscale atmospheric models (≥ 4-km resolution). However, the effect can play a crucial role in meso-urban atmospheric models as well as microscale urban dispersion models with resolutions of 1 m – 1 km.

**Acknowledgments.** This research was supported by the NSF under grant AGS-0946315 and AGS-1523296. LRL was supported by Department of Energy Office of Science Biological and Environmental Research through the Regional and Global Climate Modeling program. PNNL is operated for DOE by Battelle Memorial Institute under contract DE-AC05-76RL01830.

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



**Tables and figures**

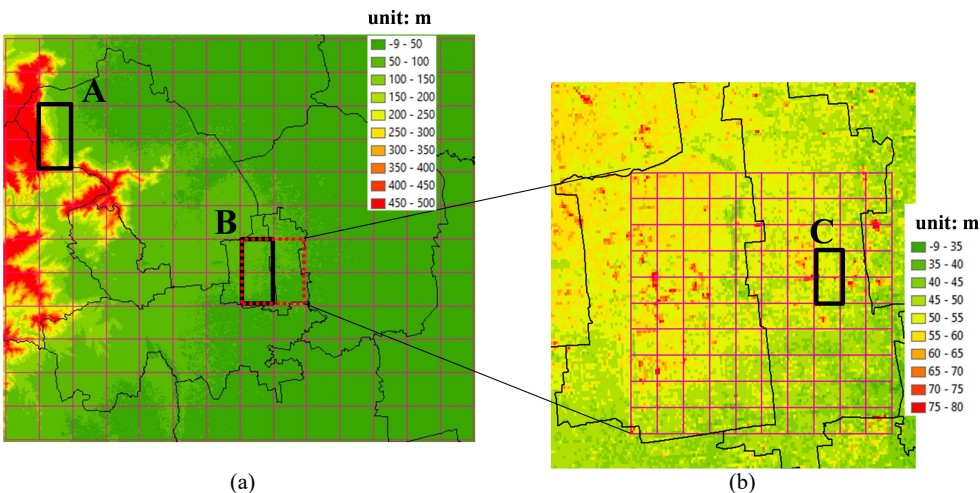

(a)              (b)

Figure 1. Modelling domains used in 3-D radiative transfer calculation: (a) Domain 1
covering urban and suburban Beijing at a grid resolution of 4 km; (b) Domain 2 covering the
urban center of Beijing at a grid resolution of 800 m. The colours represent altitudes at a
resolution of 3 arc-second (about 90 m) derived from SRTM. The black thin lines represent
boundaries of districts. The three black bold rectangles (defined as A, B, and C, respectively)
represent typical grids used to analyze diurnal variation and to quantify the contribution of
flux components. The red dashed rectangle represents grids in Domain 1 that correspond to
Domain 2.









Figure 2. Surface solar flux deviations between the 3-D radiation parameterization and the
plane-parallel scheme at selected times (7:00, 12:00, and 17:00 BT) on April 1$^{st}$ in conditions
with and without aerosols.

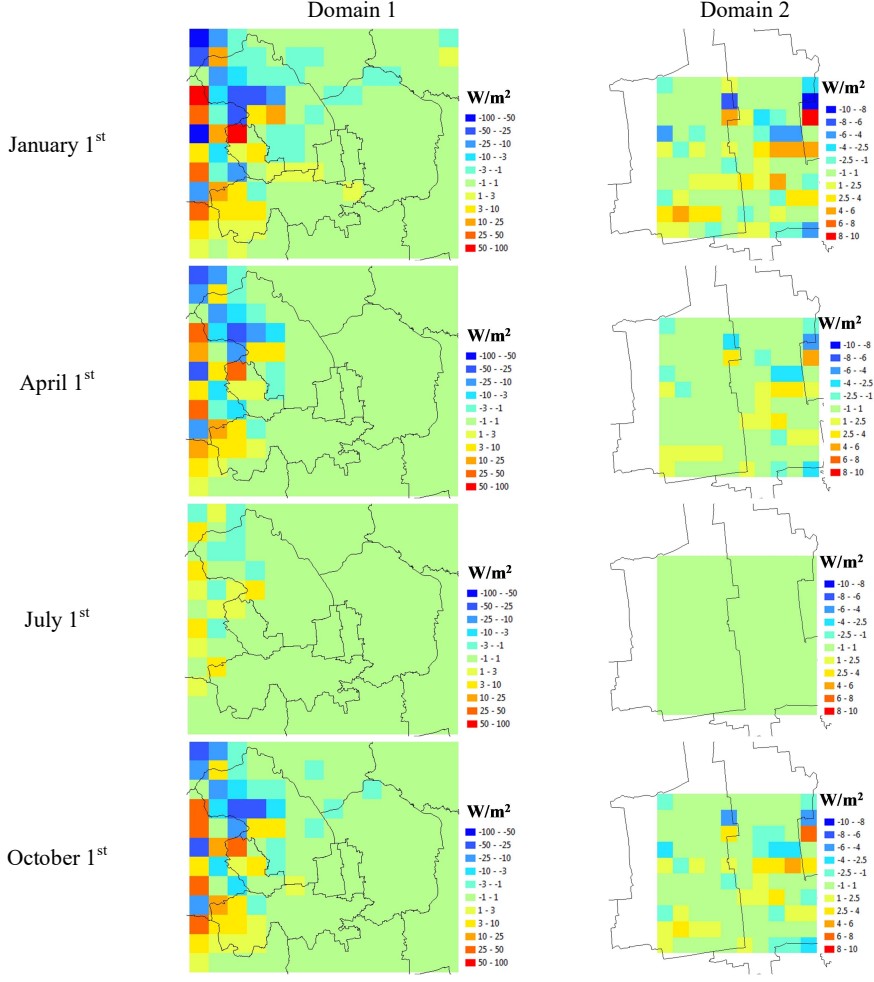

Figure 3. Daily average surface solar flux deviations between the 3-D radiation
parameterization and the plane-parallel scheme in clear-sky condition without aerosols on
January 1$^{st}$, April 1$^{st}$, July 1$^{st}$, and October 1$^{st}$, 2012.





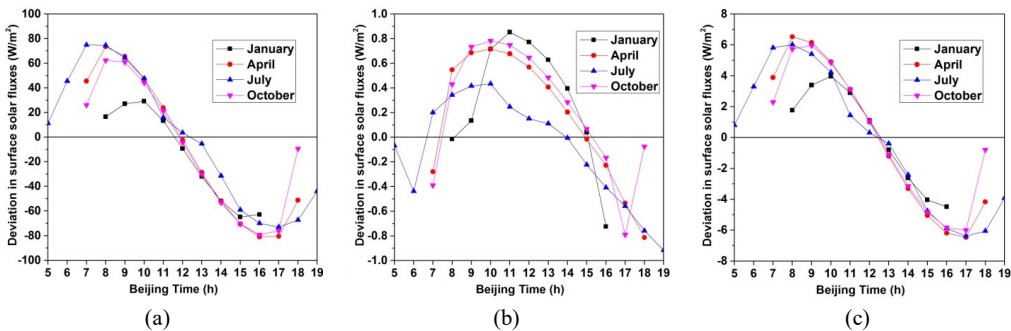

|  |  |  |
|---|---|---|
| (a) | (b) | (c) |

Figure 4. Diurnal variation of surface solar flux deviations between the 3-D radiation
parameterization and the plane-parallel scheme in clear-sky condition without aerosols in
typical grids marked by black bold rectangles in Fig. 1: (a) a typical mountainous area,
defined as rectangle A; (b) a typical urban area in Domain 1, defined as rectangle B; (c) a
typical urban area in Domain 2, defined as rectangle C.





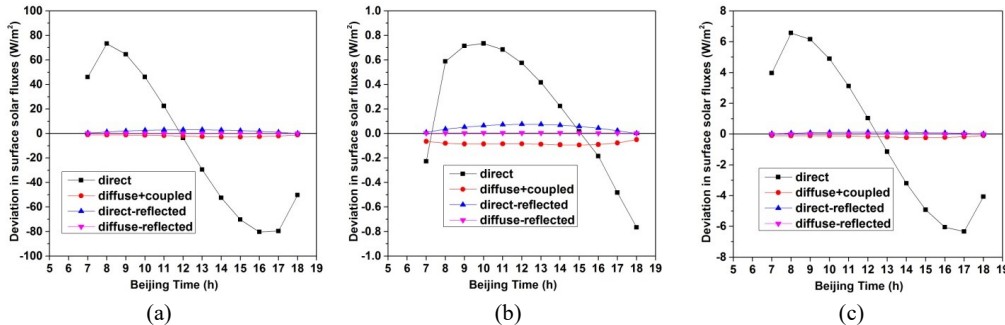

(a)                    (b)                    (c)

Figure 5. Contributions of individual components to surface solar flux deviations between the
3-D radiation parameterization and the plane-parallel scheme in clear-sky condition without
aerosols in typical grids on April 1$^{st}$. Panel (a), (b), and (c) are for the same grids as Fig. 4(a),
Fig. 4(b), and Fig. 4(c).





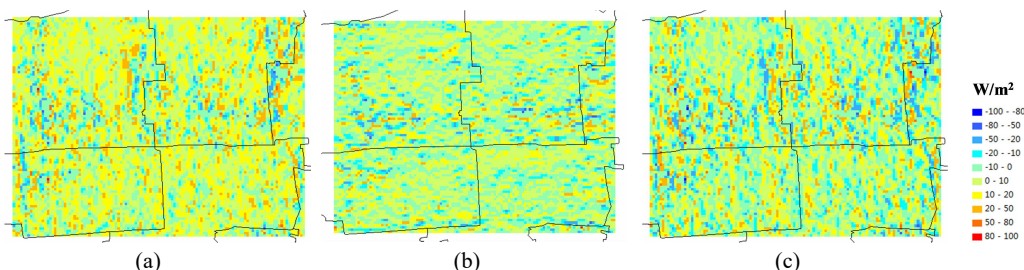

Figure 6. Surface solar flux deviations between the 3-D radiation parameterization and the
plane-parallel scheme on April 1$^{st}$ at a grid resolution of 3 arc-second (about 90 m). (a) 7:00
BT; (b) 12:00 BT; (c) 17:00 BT. The size of the simulation domain is the same as Domain 2
defined in Fig. 1.





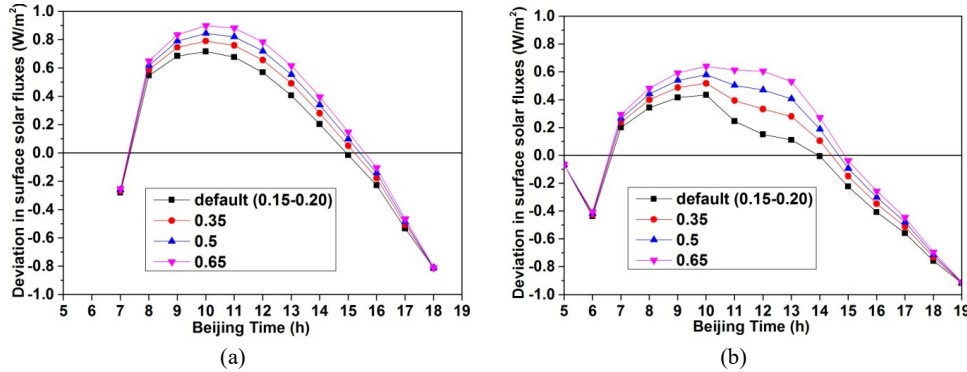

|     |     |
| :-: | :-: |
| (a) | (b) |

1  Figure 7. Sensitivity of surface solar flux deviations between the 3-D radiation

2  parameterization and the plane-parallel scheme to the surface albedo in a typical urban area in

3  Domain 1 (defined as rectangle B in Fig. 1) on (a) April 1st, (b) July 1st.