# Peer review of "Impact of buildings on surface solar radiation over urban"

_Atmospheric Chemistry and Physics, 2016_

## Referee Comment (RC1) · Anonymous Referee #3 · 3 Feb 2016

It is a pleasure to review the manuscript "Impact of buildings on surface solar radiation over urban Beijing" by Zhao et al. This manuscript addresses a challenging problem: How to take into account building effects for calculating downward solar radiation in complex urban areas, which is critical for estimating urban heat islands and for urban modeling. It examines the building effects on solar radiation by comparing, at different spatial resolutions (4 km and 800-m), results from a 3-D radiation parameterization with results from a relatively simple plane-parallel approach. The paper is well organized and well written, and the research quality is high. I suggest to accepting this manuscript after the authors clarify the following points:

- The deviations of radiation fluxes calculated from the above two models are remarkably small ($\sim$1–10 W m$-$2 at 800-m grid resolution and 1 W m$-$2 at 4-km resolution) from mesoscale modeling perspectives. Can the authors elaborate on the types of

urban data are used in their 3-D model? Judging from Fig. 1, I understand that the SRTM data provides information regarding the average building height and perhaps gaps between buildings. Does this dataset also describe building width? A more general question is: does this 3-D radiation model take into account the building geometry? If not, what would be uncertainties of not account it in the calculated deviations?

- Section 3.1, page 2, Line 18, "these grids are": which grids are referring to here? It may be useful to add a figure to show the geometry features of grids, which is not obvious from Fig. 1. - Section 3.2, Line 17: authors mentioned sky view factor (SVF). Does the 3-D model consider SVF and if yes, how? Does the calculation of SVF involve the use of building geometry characteristics?

- Section 3.3.1, Line 6: should "are now shown" be "are not shown"?

- Suggest including a plot of SRTM heights at 90-m resolution in Fig. 6.

- Section 3.3.2: authors briefly mentioned the limit of applying their models for high-resolution < 1-km? Can they elaborate more on the theoretical basis of such limitation (with reference perhaps)? It is an important issue because their models are used in this manuscript for 800-m resolution too.

- Section 3.3.3: albedo 0.15-0.2 from WRF. These albedo values likely are output of Pleim-Xue land model. Does that land model consider building effects on solar radiation (even in a crude way)? If yes, would the use of these albedo values in the 3-D model somewhat double account for building effects?

- WRF includes urban canopy models (Chen et al. 2011, International Journal of Climatology), which consider building geometry features to calculate sky view factor etc to determine effects of shadows and reflection on solar radiation. Detailed description of these models and their treatments of effects of buildings on solar radiation can be found in Kusaka and Kimura (2004, Journal of the Meteorological Society of Japan) and in Martilli and Rotach (2002, Boundary-Layer Meteorology). How does the 3-D
model differ from these simplified building-radiation schemes in WRF? I think it would be useful to compare the 3-D model results with these calculated by WRF-urban at 1-4 km scales, which are commonly used in mesoscale models for urban areas.

---

## Referee Comment (RC2) · Anonymous Referee #2 · 28 Feb 2016

This study systematically examines the impact of buildings on downward surface solar fluxes over Beijing by using a 3-D radiation parameterization that accounts for 3-D building structures versus the conventional plane-parallel scheme. Results show that the downward surface solar flux deviations between the two schemes are $\pm 1$–10 W m-2 at 800-m grid resolution. However, flux deviations are much smaller at 4-km resolution because the pairs of positive-negative flux deviations on different sides of buildings offset each other. Diurnal variations of flux deviations and contribution of individual flux components (e.g. direct flux, diffuse flux, etc.) are also analyzed. Further sensitivity experiments show that atmospheric aerosols can evidently reduce the magnitude of flux deviations while the surface albedo generally has a rather moderate impact. The results imply that the building effect on downward surface solar fluxes can play a crucial role in fine-resolution atmospheric models with grid spacing of 1 m – 1 km.

[Figure]

The subject is interesting, the paper is well written and the results are useful for urban-scale and meso-scale modeling applications. In fact I have two minor comments only:

1. The authors mainly compare the surface downward solar fluxes simulated by 3-D radiation parameterization and plane-parallel schemes. What about the difference between those simulated by 3-D radiation parameterization and by single- or multiple-layer urban canopy scheme? Some discussion about this would be helpful given that urban canopy schemes are widely used in urban climate applications.

2. In the 3-D radiation parameterization scheme, solar radiative fluxes can be categorized into five components (i.e. direct flux, diffuse flux, and so on) according to photon path. I am wondering whether it is possible to partition the total flux into individual components at roof, wall, and road, which are variables that can be used to calculate the canopy temperature and overall energy exchange between urban surface and atmosphere.

---

## Author Comment (AC1) · 14 Apr 2016

Please find the response as well as the revised manuscript in the attachment. Thank you!

Please also note the supplement to this comment: http://www.atmos-chem-phys-discuss.net/acp-2016-3/acp-2016-3-AC1-supplement.zip

---

## Author Response (AR1)

**Reviewer 1:**

It is a pleasure to review the manuscript "Impact of buildings on surface solar radiation over urban Beijing" by Zhao et al. This manuscript addresses a challenging problem: How to take into account building effects for calculating downward solar radiation in complex urban areas, which is critical for estimating urban heat islands and for urban modeling. It examines the building effects on solar radiation by comparing, at different spatial resolutions (4 km and 800-m), results from a 3-D radiation parameterization with results from a relatively simple plane-parallel approach. The paper is well organized and well written, and the research quality is high. I suggest to accepting this manuscript after the authors clarify the following points:

Response: We would like to thank the reviewer for kind words and appreciate his/her constructive comments to improve the quality of our manuscript. In the following, the original comments are in black, while our responses are in blue.

- The deviations of radiation fluxes calculated from the above two models are remarkably small (~1-10 W m-2 at 800-m grid resolution and 1 W m-2 at 4-km resolution) from mesoscale modeling perspectives. Can the authors elaborate on the types of urban data are used in their 3-D model? Judging from Fig. 1, I understand that the SRTM data provides information regarding the average building height and perhaps gaps between buildings. Does this dataset also describe building width? A more general question is: does this 3-D radiation model take into account the building geometry? If not, what would be uncertainties of not account it in the calculated deviations?

Response: We appreciate the reviewer's valuable comment. In the 3-D model, we use topography data at a resolution of 3 arc-second (about 90 m) from the Shuttle Radar Topography Mission (SRTM), which provides the elevation of each 90-m pixel. In other words, the SRTM data mimic buildings as numerous $90{\times}90$ m$^2$ cuboids and define the width of large buildings comprising of a number of 90-m pixels. However, the detailed geometry of buildings is not resolved, representing a limitation of this study, which could introduce uncertainty in evaluating the building effect on solar fluxes.

The quantification of this uncertainty is beyond the scope of this study – a subject requiring further investigations. Nevertheless, we present a brief discussion in the following.

The grid resolutions used in our 3-D radiation simulation (4 km and 800 m) are considerably larger than most buildings. We found that positive and negative flux deviations on different sides of buildings often offset each other within a model grid. For this reason, the detailed geometry of buildings produces relatively minor effects on simulated solar fluxes. The effects could be much larger if we would use a grid resolution comparable to the dimension of most buildings, which is, however, beyond the scope of this study.

We have also described the urban topography data used and their limitation in the revised manuscript (Page 5 Line 17-21, and Page 14 Line 14-23 of the revised manuscript).

- Section 3.1, page 2, Line 18, "these grids are": which grids are referring to here? It may be useful to add a figure to show the geometry features of grids, which is not obvious from Fig. 1.

Response: We thank the reviewer for this comment. We have revised this sentence as follows to make it clearer.

Figure 1 shows that the typical urban area defined in Domain 2 (i.e., rectangle C) is mostly located in the eastern side of the buildings rather than in the western side. (Page 8 Line 27-29 of the revised manuscript)

As described above, the SRTM topography data used in this study cannot resolve the detailed geometry of buildings, but only provide the elevation of each 90-m pixel, as shown in Figure 1.

- Section 3.2, Line 17: authors mentioned sky view factor (SVF). Does the 3-D model consider SVF and if yes, how? Does the calculation of SVF involve the use of building geometry characteristics?

Response: We thank the reviewer for this valuable comment. The 3-D model does consider sky view factor when quantifying the building effect on solar radiation. The 3-D model parameterizes solar flux deviations from horizontal surface as a function of grid-average topographic parameters. Sky view factor is one of these topographic parameters. The governing equation of the 3-D model is shown as follows:

$$
\begin{pmatrix} F_{dir}^{'} \\ F_{dif}^{'} \\ F_{rdir}^{'} \\ F_{rdif}^{'} \\ F_{coup}^{'} \end{pmatrix} = \begin{pmatrix} a_1 \\ a_2 \\ a_3 \\ a_4 \\ a_5 \end{pmatrix} + \begin{pmatrix} b_{11} & b_{12} & 0 & 0 \\ b_{21} & b_{22} & 0 & b_{24} \\ 0 & b_{32} & b_{33} & 0 \\ 0 & b_{42} & b_{43} & 0 \\ b_{51} & b_{52} & b_{53} & 0 \end{pmatrix} \begin{pmatrix} \langle \tilde{\mu}_i \rangle \\ \langle \tilde{V}_d \rangle \\ \langle \tilde{C}_t \rangle \\ \sigma(h) \end{pmatrix} \tag{1}
$$

where $F_i^{'}$ is the relative deviation of each flux component, i = dir (direct flux), dif (diffuse flux), rdir (direct-reflected flux), rdif (diffuse-reflected flux), and coup (coupled flux). $a_i$ is the interception, $b_{ij}$ is the regression coefficient for a specific independent variable. $\tilde{\mu}_i$ is the cosine of the solar zenith angle normalized by the cosine of the slope, $\tilde{V}_d$ is the sky view factor normalized by the cosine of the slope, $\tilde{C}_t$ is the terrain configuration factor normalized by the cosine of the slope, $\sigma(h)$ is the standard deviation of elevation, and angle brackets denote the spatial mean of the variable within a model grid.

The sky view factor is calculated using the SRTM elevation data at ~90 m resolution. The overall heights, widths, and intervals of buildings are defined in the SRTM data and thus used in sky view factor calculation, but the detailed building geometry is not accounted for. We first calculate the sky view factor for each 90-m pixel, followed by evaluating the average sky view factor for each 4-km or 800-m grid in simulation domains.

If one assumes that the diffuse flux is isotropic, the sky view factor represents the portion of the sky dome visible to a target point. The latter can be obtained by calculating the area of unobstructed hemisphere given by

$$V_d = \frac{1}{\pi} \int_0^{2\pi} \int_0^{H_\varphi} \sin\theta \left[ \cos\theta \cos\theta_s + \sin\theta \sin\theta_s \cos(\varphi - \varphi_s) \right] d\theta d\varphi$$

$$\approx \frac{1}{2\pi} \int_0^{2\pi} \left[ \cos\theta_s \sin^2 H_\varphi + \sin\theta_s \cos(\varphi - \varphi_s) \times \left( H_\varphi - \sin H_\varphi \cos H_\varphi \right) \right] d\varphi \qquad (2)$$

where $\theta_s$ is the slope, $\varphi_s$ is the aspect of the slope, and $H_\varphi$ is the horizon angle measured from the zenith down to the local horizon at the orientation direction $\varphi$, which can result either from "self‐shadowing" by the slope itself or from nearby buildings. For more details about the calculation of sky view factor, please refer to our previous paper (Lee et al., 2011).

We have given these descriptions in the revised manuscript (from Page 4 Line 10 to Page 5 Line 4, and Page 5 Line 17-24 of the revised manuscript).

Reference:

Lee, W. L., Liou, K. N., and Hall, A.: Parameterization of solar fluxes over mountain surfaces for application to climate models, J Geophys Res-Atmos, 116, D01101, DOI 10.1029/2010jd014722, 2011.

- Section 3.3.1, Line 6: should "are now shown" be "are not shown"?

Response: We apologize for this mistake. We have changed "are now shown" to "are not shown".

- Suggest including a plot of SRTM heights at 90-m resolution in Fig. 6.

Response: We appreciate the reviewer's comment. In fact, we have provided a plot of SRTM heights at ~90-m resolution in the right panel of Figure 1. To make it clearer, we added a sentence in the caption of Figure 6 which refers the reader to Figure 1 for the elevation map. The revised caption of Figure 6 is shown as follows.

Figure 6. Surface solar flux deviations between the 3-D radiation parameterization and the plane-parallel scheme on April 1st at a grid resolution of 3 arc-second (about 90 m). (a) 7:00 BT; (b) 12:00 BT; (c) 17:00 BT. The size of the simulation domain is the same as Domain 2 defined in Fig. 1. An elevation map at a 90-m resolution for the simulation domain is also shown in the right panel of Fig. 1.

- Section 3.3.2: authors briefly mentioned the limit of applying their models for high resolution < 1-km? Can they elaborate more on the theoretical basis of such limitation (with reference perhaps)? It is an important issue because their models are used in this manuscript for 800-m resolution too.

Response: We appreciate the reviewer's valuable comment. We mentioned that the parameterization may not be applicable to a fine resolution < about 1 km with good accuracy (Lee et al., 2013). Here "about 1 km" is not a strict limit but rather a rough estimate. The theoretical basis is that the 3-D radiation parameterization is based on grid-average topographic parameters, so it is valid only if a model grid comprises a number of topographic pixels (Lee et al., 2011, 2013). The present parameterization is applicable to a grid resolution as fine as 800 m since an 800×800 m$^2$ grid encompasses a large number of 90-m pixels. In Section 2.2, we demonstrated the applicability of an 800-m resolution by comparing flux deviations in each 4×4 km$^2$ grid calculated directly from the 3-D parameterization and those from the summation of all 800×800 m$^2$ grids. (from Page 5 Line 28 to Page 6 Line 5, and Page 11 Line 2-6 of the revised manuscript).

Response: We appreciate the reviewer's valuable comment. The 3-D radiation parameterization used in this study and the urban canopy models used in many previous urban climate studies investigate the building effect on solar radiation with different methods. The present 3-D radiation parameterization was developed on the basis of a 3-D Monte Carlo photon tracing approach. We parameterize the downward surface solar fluxes using the grid-average topographic information to reproduce the results in order to reduce the computational burden of 3-D Monte Carlo photon tracing calculations. In urban canopy models, the radiative transfer equation was established by employing simplified, evenly spaced buildings. Then, the average building geometry parameters (e.g., building height, building width, street width, etc.) are calculated for each model grid and subsequently used in radiative transfer calculation (Chen et al. 2011; Grimmond et al., 2010; Kusaka and Kimura, 2004; Martilli et al., 2002).

The present 3-D radiation parameterization and the commonly used urban canopy models both have their advantages and limitations. The 3-D radiation parameterization directly relates surface solar fluxes to "real" (rather than simplified) topographic data, resulting in the realistic spatiotemporal distribution of solar fluxes. In addition, the 3-D radiation parameterization treats individual flux components (i.e., the direct, diffuse, direct-reflected, diffuse-reflected, and coupled fluxes) separately taking into account the distinct feature of different flux components. However, the diffuse, diffuse-reflected, and coupled fluxes are usually oversimplified in urban canopy models (e.g. isotropic radiation is assumed). Some advantages of urban canopy models include a more detailed treatment of building geometrical features as well as the partitioning of total fluxes at roof, wall, and road to facilitate computations of the energy exchange at building domain.

We fully agree with the reviewer that it would be useful to compare the present 3-D model results with those calculated by urban canopy models. However, such a comparison would involve substantial and collaborative efforts, resources, and coordination with scientists who have already developed a successful urban canopy model. We will be pleased to work on a future paper if the reviewer could be so kind as to direct us to the opportunity for collaborative work.

We have added the discussions above in the revised manuscript (from Page 13 Line 22 to Page 14 Line 26 of the revised manuscript).

Response: We appreciate the reviewer's valuable comment. The 3-D radiation parameterization used in this study and urban canopy schemes used in many previous urban climate studies investigate the building effect on solar radiation with different methods. The present 3-D radiation parameterization was developed on the basis of a 3-D Monte Carlo photon tracing approach through which we parameterize downward surface solar fluxes by means of grid-average topographic information to reproduce flux results in order to optimize the computational burden involving 3-D Monte Carlo photon tracing calculations. In urban canopy schemes, the radiative transfer equation was first established using simplified, evenly spaced buildings. The average building geometry parameters (e.g., building height, building width, street width, etc.) are subsequently calculated for each model grid and used in radiative transfer calculations (Chen et al. 2011; Grimmond et al., 2010; Kusaka and Kimura, 2004; Martilli et al., 2002).

The present 3-D radiation parameterization and the commonly used urban canopy schemes both have their strengths. The 3-D radiation parameterization treats individual flux components (i.e., the direct, diffuse, direct-reflected, diffuse-reflected, and coupled fluxes) separately taking into account the distinct feature of different flux components. However, the diffuse, diffuse-reflected, and coupled fluxes are usually oversimplified in urban canopy schemes (e.g. isotropic radiation is assumed). Besides, the 3-D radiation parameterization directly relates surface solar fluxes to "real" (rather than simplified) topographic data, resulting in the realistic spatiotemporal distribution of solar fluxes. Some advantages of urban canopy schemes include a more detailed treatment of building geometrical features as well as the partitioning of total fluxes at roof, wall, and road to facilitate computations of the energy exchange at building domain.

It would be interesting to compare solar fluxes simulated by 3-D radiation parameterization and by single- or multiple layer urban canopy schemes. Such a comparison, however, would involve substantial and collaborative efforts, resources, as well as the coordination with scientists who have already developed an urban canopy scheme. We submit that such a comparison appears to be beyond the purview of the present study. We will be pleased to work on a future paper if the reviewer could enlighten us to the opportunity for collaborative work.

We have added the discussions above in the revised manuscript (from Page 13 Line 22 to Page 14 Line 26 of the revised manuscript).

Response: We thank the reviewer for this valuable comment. For each model grid, the 3-D radiation parameterization used in this study can partition total fluxes into five components (i.e., the direct, diffuse, direct-reflected, diffuse-reflected, and coupled fluxes) according to photon path. However, it is difficult to partition total fluxes into components at roof, wall, and road since the 3-D radiation parameterization does not resolve complex building geometrical parameters. If the roof or road occupies a model grid, then it's possible to get different components of the fluxes for roof or road; otherwise the partitioning will be very difficult. We have described this limitation in the revised manuscript (Page 14 Line 23-26 of the revised manuscript).

[revised manuscript text omitted]